# The Lysine Demethylases KdmA and KdmB Differently Regulate Asexual Development, Stress Response, and Virulence in *Aspergillus fumigatus*

**DOI:** 10.3390/jof8060590

**Published:** 2022-05-31

**Authors:** Yong-Ho Choi, Min-Woo Lee, Kwang-Soo Shin

**Affiliations:** 1Department of Microbiology, Graduate School, Daejeon University, Daejeon 34520, Korea; yougnho1107@gmail.com; 2Soonchunhyang Institute of Medi-Bio Science, Soonchunhyang University, Cheonan 31151, Korea

**Keywords:** *Aspergillus fumigatus*, lysine demethylase, transcriptomics, oxidative stress, 6-azauracil, gliotoxin, virulence

## Abstract

Histone demethylases govern diverse cellular processes, including growth, development, and secondary metabolism. In the present study, we investigated the functions of two lysine demethylases, KdmA and KdmB, in the opportunistic human pathogenic fungus *Aspergillus fumigatus*. Experiments with mutants harboring deletions of genes encoding KdmA (Δ*kdmA*) and KdmB (Δ*kdmB*) showed that KdmA is necessary for normal growth and proper conidiation, whereas KdmB negatively regulates vegetative growth and conidiation. In both mutant strains, tolerance to H_2_O_2_ was significantly decreased, and the activities of both conidia-specific catalase (CatA) and mycelia-specific catalase (Cat1) were decreased. Both mutants had significantly increased sensitivity to the guanine nucleotide synthesis inhibitor 6-azauracil (6AU). The Δ*kdmA* mutant produced more gliotoxin (GT), but the virulence was not changed significantly in immunocompromised mice. In contrast, the production of GT and virulence were markedly reduced by the loss of *kdmB*. Comparative transcriptomic analyses revealed that the expression levels of developmental process-related genes and antioxidant activity-related genes were downregulated in both mutants. Taken together, we concluded that KdmA and KdmB have opposite roles in vegetative growth, asexual sporulation, and GT production. However, the two proteins were equally important for the development of resistance to 6AU.

## 1. Introduction

Chromatin is a highly dynamic structure that underpins numerous eukaryotic nuclear processes, such as transcription, DNA replication, recombination, and repair. Nucleosomes are fundamental units of chromatin structure, and they undergo remodeling as a result of post-translational modifications placed on or removed from histones present in the nucleosome [1,2,3,4]. Histones are core nucleosome proteins that form a histone octamer with two dimers of H2A–H2B and a tetramer of H3–H4, around which DNA is wound. Each core histone has an N-terminal amino acid tail extending from the nucleosome, which can be subjected to many modifications, which, in turn, influence chromatin structure and function. Since the discovery that histone-modifying enzymes alter secondary metabolite (SM) production [5], the functions of these enzymes in filamentous fungi have been intensively studied [6,7,8].

The lysine-specific histone demethylase KDM1A/LSD1 removes mono- and dimethyl groups from H3K4 using its amine oxidase domain in a flavin adenine dinucleotide-dependent manner. Another histone demethylase, KDM2A, removes methyl groups from H3K36 using its Jumonji C (JmjC) domain, which catalyzes lysine demethylation in histones through an oxidative reaction that requires Fe (II) and α-ketoglutarate as cofactors [9,10]. To date, 20 histone demethylases have been identified, characterized, and divided into the lysine-specific histone demethylase (LSD) family or the JmjC domain-containing family [11]. LSD demethylases can only demethylate mono- and dimethylated histone lysine residues, whereas JmjC domain family demethylases remove methyl groups from lysine residues in all three methylation states.

In *Aspergillus nidulans*, the *kdmA* encodes KdmA, a member of the KDM4 family of Jumonji histone demethylase proteins that demethylates H3K9 and H3K36 [7]. KdmA plays a particularly important role in H3K36me3 demethylation and exerts both positive and negative effects on gene expression [7]. According to recent transcriptome analysis, about 25–30% of the genome is under the influence of KdmA, which acts as a transcriptional repressor of primary metabolism genes and activator of secondary metabolism [7]. Another Jumonji histone demethylase, KdmB (*A. nidulans* KDM5 homolog), which belongs to the JARID group of enzymes, was recently discovered [12]. Members of the Jumonji and AT-Rich Interaction Domain (JARID) subfamily of demethylases target di- and trimethylated H3K4. Because high levels of H3K4 methylation are detected in the promoter regions of active genes [13], KdmB is thought to repress gene transcription by removing the active mark of H3K4, thereby downregulating the transcription. Functional characterization of KdmB revealed that several genes were upregulated by its deletion [12]. This indicates that KdmB functions as a transcriptional repressor by catalyzing H3K4 demethylation. Furthermore, KdmB is required for the full expression of several genes involved in the regulation of SM. KdmB is a multi-domain protein consisting of the JmjC, JmjN, ARID/BRIGHT, and zinc finger domains, as well as two plant homeodomains (PHDs). The JmjC domain contains binding sites for Fe (II) and α-ketoglutarate and is required for the catalytic demethylase activity. The role of the JmjN domain remains unclear, but it is thought to support the catalytic function of JmjC [14,15]. Although the ARID/BRIGHT DNA-binding domain, PHD, and zinc finger domain are required for the demethylation activity of JARID1 proteins [15], their functions are not yet completely understood.

Previous studies showed that histone demethylases in filamentous fungi are greatly involved in regulating secondary metabolism. Deletion of *kdmA* and *kdmB* altered the SM profile in *A. nidulans* [7,12]. However, the precise functions of histone demethylases in the human pathogenic fungus *A. fumigatus* remained unknown. In this study, we investigated the characteristics and functions of *kdmA* and *kdmB*, which encode histone demethylases KdmA and KdmB, and show that they differently regulate asexual development, stress response, toxin production, and virulence in *A. fumigatus*.

## 2. Materials and Methods

### 2.1. Bioinformatic Analysis of KdmA and KdmB

The amino acid sequences of *A*. *nidulans kdmA* (AN1060) and *kdmB* (AN8211) were used to search homologs in the *A. fumigatus* Af293 genomic database (http://www.aspergillusgenome.org/, accessed on 2 December 2020). Amino acid sequences of the full length of KdmA and KdmB were aligned with the SMART program (http://smart.embl-heidelberg.de, accessed on 4 April 2021) for structural comparison, and the JmjC domain of demethylase homologs was aligned by MEGA7 software (http://www.megasoftware.net, accessed on 20 July 2021) for phylogenetic analysis.

### 2.2. Strains and Culture Conditions

All *A. fumigatus* strains used in this study (Table 1) were derivatives of the WT Af293 strain [16]. Fungal strains were grown on a minimal medium with glucose (MMG) or MMG with 0.1% yeast extract (MMY) and appropriate supplements, as described previously [17]. For RNA-seq analysis, porous cellophane was deposited on the surface of the agar, point-inoculated with approximately 1 × 10^5^ conidia onto the cellophane, and incubated in the dark at 37 °C for 3 days. Fungal tissue was collected using a sterile spatula and used for RNA extraction. For Western blotting, nuclei of each strain were prepared as the previously described method [18]. Briefly, cultures (500 mL) of MMG were inoculated with 1 × 10^6^ conidia/mL and incubated at 37 °C for 48 h. Mycelia were collected by vacuum filtration, frozen in liquid nitrogen, and ground to powder under liquid nitrogen. Ground mycelial powder was suspended in 200 mL nuclei isolation buffer (1 M sorbitol, 10 mM Tris-HCl, pH 7.5, 0.15 mM spermine, 0.5 mM spermidine, 10 mM EDTA, 2.5 mM phenylmethylsulfonyl fluoride) on ice. Samples were centrifuged, and the supernatant was filtered through Miracloth. The filtrate was centrifuged, the pellet was resuspended in 15 mL resuspension buffer (nuclei isolation buffer containing 1 mM EDTA) and then centrifuged, and the supernatant was removed, and crude nuclei were resuspended in 0.6 mL of ST buffer (1 M sorbitol, 10 mM Tris-HCl, pH 7.5, and protease inhibitor cocktail). Debris was pelleted by centrifugation, and the protein concentration was determined in a Bradford assay in triplicate. For stress experiments, 200 mL of liquid YG cultures were inoculated with 1 × 10^5^ conidia/mL and grown at 37 °C with 200 rpm. After 18 h, each chemical (5 mM H_2_O_2_, 200 μM paraquat, 100 μg/mL 6-azauracil) was added and further incubated for 24 h.

### 2.3. Construction of Mutant A. fumigatus Strains

The deletion construct generated by double-joint PCR [20] containing the *A. nidulans* selective marker (*AnipyrG*) with the 5′ and 3′ flanking regions of the *A*. *fumigatus kdmA* gene (Afu1g12332) and *kdmB* gene (Afu5g03430) was introduced into the recipient strains [21]. The selective marker was amplified from *A. nidulans* FGSC A4 genomic DNA using primer pair oligo697/oligo698. The null mutant colonies were isolated and confirmed by diagnostic PCR (using primer pairs oligo1400/oligo1401 and oligo1406/oligo1407) followed by restriction enzyme digestion. In order to complement the mutants, a double-joint PCR method was used [20] with *hygB* as a selective marker. The oligonucleotides used in this study are listed in Appendix A.

### 2.4. Nucleic Acid Isolation and Manipulation

Total RNA isolation and RT-qPCR were performed as previously described [22,23,24]. Briefly, each sample was homogenized in 1 mL of TRIzol reagent (Invitrogen, Waltham, MA, USA), and the supernatant was mixed with an equal volume of ice-cold isopropanol and centrifuged. The RNA pellets were washed with 70% ethanol using diethyl pyrocarbonate-treated water and dissolved in the RNase-free water. RT-qPCR was performed using a Rotor-Gene Q instrument (Qiagen, Hilden, Germany). Each run was assayed in triplicate in a total volume of 20 µL containing the RNA template, One-Step RT-PCR SYBR Mix (Doctor Protein, Seoul, Korea), reverse transcriptase, and 10 pmoles of each primer. Reverse transcription was carried out at 42 °C for 30 min. PCR conditions were 95 °C/5 min for one cycle, followed by 95 °C and 55 °C/30 s for 40 cycles. The amplification of a specific target DNA was verified by melting curve analysis. The expression ratios were normalized to the expression level of the reference gene *ef1α* [25,26] and calculated using the ΔΔCq method [27]. The expression stability of *ef1α* and efficiencies of PCRs of the target genes were determined as previously described [28]. Expression levels of target gene mRNAs were analyzed using appropriate oligonucleotide pairs (Appendix A). For RNA-seq analyses, total RNA was extracted and submitted to eBiogen, Inc. (Seoul, Korea) for library preparation and sequencing.

### 2.5. Physiological Experiments

Radial growth and sensitivity to chromatin-targeting inhibitors were measured as described previously [18]. In order to determine the number of conidia, two inoculation methods were used; point-inoculated cultures were used as per the radial growth experiment, and overlay-inoculated cultures were used as the total conidia quantification per plate [18]. Conidia were collected using 0.5% Tween 80 solution from the plates, filtered through Miracloth (Calbiochem, San Diego, CA, USA), and counted using a hemocytometer. In order to test for oxidative stress, paraquat (PQ, 200 µM) and H_2_O_2_ (7.5 mM) were added to the YG medium after autoclaving. Conidial suspensions (1 × 10^4^ conidia) were inoculated into a solid RPMI-1640 medium, and E-test strips were placed on the plate. After incubation at 37 °C for 48 h, MIC values were determined as the zone edge intersecting the strips. The production of secondary metabolites and GT was determined as described previously [28,29]. GT standard was purchased from Sigma-Aldrich (Burlington, MA, USA).

### 2.6. Enzyme Assay and Western Blotting

In order to determine catalase and SOD activities, the conidia (1 × 10^5^) of relevant strains were inoculated into liquid YG with appropriate supplements and incubated at 37 °C, 250 rpm for 24 h. Then, oxidative stress agents were added, and the conidia were incubated for another 24 h. The mycelia were disrupted with glass beads in 20 mM phosphate buffer (pH 7.5) supplemented with a protease inhibitor cocktail. Protein content was quantified using Bradford reagent (Bio-Rad Laboratories, Inc., Hercules, CA, USA) and bovine serum albumin as a standard. Catalase activity was visualized by negative staining with ferricyanide [30]. SOD activity was detected as the inhibition of nitroblue tetrazolium reduction [31]. For Western blotting, nuclear extracts were isolated as previously described [32]. Approximately 50 µg of nuclear protein extract was electrophoresed on a 15% SDS-PAGE gel and subsequently electroblotted to nitrocellulose membranes. Relevant histone modifications were detected with primary antibodies specific to histone H3 (Abcam, 1791), H3K4me3 (Abcam, 8580), and H3K36me3 (Abcam, 9050) antibodies. Relative intensities of Western blot and the enzyme activities were quantified using the Image J 1.52k software (NIH, Bethesda, MD, USA).

### 2.7. Transcriptome Analysis

Library construction was performed using the QuantSeq 3′ mRNA-Seq Library Prep Kit (Lexogen, Inc., Wien, Austria) according to the manufacturer’s instructions. High-throughput sequencing was performed as single-end 75 sequencing using a NextSeq 500 system (Illumina, Inc., San Diego, CA, USA). QuantSeq 3′ mRNA-Seq reads were aligned using Bowtie2 [33]. The alignment file was used for assembling transcripts, estimating their abundance, and detecting the differential expression of genes. Differentially expressed genes were determined based on the counts from unique and multiple alignments using coverage in Bedtools [34]. The RT (read count) data were processed based on the quantile normalization method using EdgeR within R (R Development Core Team, 2016) using Bioconductor [35]. Gene classification was based on searches performed using DAVID (http://david.abcc.ncifcrf.gov/, (accessed on 25 February 2021)) and Medline databases (http://www.ncbi.nlm.nih.gov/, (accessed on 25 February 2021)). Hierarchical clustering was performed using ExDEGA (Excel-based Differentially Expressed Gene Analysis) software (ver. 3.0, eBiogen Inc., Seoul, Korea). The data were confirmed by RT-qPCR.

### 2.8. Murine Virulence Assay

The virulence assay was conducted as previously described [28,36]. Briefly, for the immunocompromised mouse model, we used outbred CrlOri: CD1 (ICR) (Orient Bio Inc., Seongnam, Korea) female mice (6–8 weeks old, average weighing 30 g), which were housed at a density of five per cage and provided access to food and water ad libitum. The mice were immunosuppressed by treatment of cyclophosphamide (250 mg/kg on day −3 and −1 and 125 mg/kg on day +1) and cortisone acetate (250 mg/kg on day −1 and 125 mg/kg on day +3). On day 0, mice were intranasally infected. For conidia inoculation, mice were anesthetized with isoflurane and then intranasally infected with 1 × 10^7^ conidia of *A. fumigatus* strains (10 mice per each fungal strain) suspended in 30 µL of 0.01% Tween 80 in PBS. Mice were monitored every 12 h for 8 days after the challenge. Control mice used in all experiments were inoculated with sterile 0.01% Tween 80 in PBS. For histology experiments, we sacrificed the mice 3 days after conidia infection. Kaplan–Meier survival curves were analyzed using the log-rank (Mantel–Cox) test for significance. Histological quantification of fungal burden was determined using the Image J 1.52k software (NIH, Bethesda, MD, USA) as described previously [37].

### 2.9. Statistical Analysis

All experiments were performed in triplicate, and *p* < 0.05 was considered a significant difference. GraphPad Prism 4 (GraphPad Software, Inc., San Diego, CA, USA) was used for the statistical analyses and graphical presentation of the survival curve.

## 3. Results

### 3.1. Structure of KdmA and KdmB of A. fumigatus

Based on the domain composition, the full-length KdmA of *A. fumigatus* (KdmA, Afu1g12332) is classified as a member of the mammalian KDM4 family of proteins (JHDM3/JMJD2) and an ortholog of *A. nidulans* KdmA (AN1060). Proteins belonging to this family demethylate di- and trimethylated lysine, including lysine 9 and 36 of histone H3 [38,39,40,41,42]. The domain architecture of KdmA is similar to that of orthologs from other species that contain the N-terminal JmjN, JmjC, and chromatin-binding PHD domains (Figure 1A). Domain analysis revealed that KdmB (Afu5g03430, an ortholog of *A. nidulans* KdmB) is a JARID1 family histone H3 lysine demethylase. In addition to the JmjC and JmjN domains, KdmB harbors an ARID/Bright domain (aa 160–257), a C5HC2 zinc finger motif (aa 861–920), a PLU-1 domain (aa 933–1276), and a PHD domain (aa 1337–1382) at the C-terminus, which are absent from KdmA homologs (Figure 1A). The amino acid sequences of the catalytic JmjC domains of histone demethylases KdmA and KdmB showed 95.7% and 94.0% identity with KdmA and KdmB of *A. nidulans*, respectively. The amino acid sequence identities of KdmA and KdmB with other *Aspergillus* orthologs were in the range from 92.6% (KdmA of *A. oryzae* and *A. flavus*) to 97.0% (KdmB of *A. oryzae* and *A. flavus*). Conserved amino acids responsible for binding cofactors Fe (II) and α-ketoglutarate [9,43] were present in all tested proteins (Figure 1B). In the unrooted phylogenetic analysis based on the amino acid sequences of the JmjC domain, each KdmA and KdmB ortholog of *Aspergillus* was clustered in the same group (Figure 1C). It was reported that KdmA functioned as H3K36 demethylase and KdmB targeted and demethylated H3K4 in *A. nidulans* [7,12]. Therefore, we performed Western blot analysis as the same substrates, and we found that KdmB demethylates H3K4 (Figure 1D).

### 3.2. KdmA and KdmB Influence Vegetative Growth and Asexual Sporulation

In order to investigate the functions of KdmA and KdmB, their respective encoding genes, *kdmA* and *kdmB*, were deleted in *A. fumigatus*, and the resulting phenotypes were monitored. Whereas the colony diameter of the mutant with *kdmA* deletion (Δ*kdmA*) was considerably lower than that of the WT strain in the solid glucose minimal medium with 0.1% yeast extract (MMY), yeast extract glucose (YG), or potato dextrose agar (PDA) media (Figure 2A). We also generated relevant complemented strains and found that their phenotypes in culture were similar to those of the WT strain (Appendix A). Additionally, in the MMY medium, quantitative analyses of the number of conidia per plate revealed that asexual spore production in the Δ*kdmA* mutant was significantly decreased to approximately 41% of that in the WT (2.0 × 10^7^ conidia). In contrast, in the Δ*kdmB* mutant, the number of conidia was slightly increased to approximately 116% of that in the WT (Figure 2B). Similarly, the number of conidia per growth area, which was determined with point-inoculated cultures, was also significantly decreased in the Δ*kdmA* mutant (1.5 × 10^5^ conidia/cm^2^, about 76% of WT strain, data not shown). In accordance with this observation, mRNA expression levels of the key asexual development regulators *abaA*, *brlA*, and *wetA* were altered in different ways in the two mutants (Figure 2C).

### 3.3. KdmA and KdmB Affect Response to Oxidative Stress

Transcriptomic analysis revealed that Δ*kdmA* and Δ*kdmB* mutants responded differently to external oxidative stress. As shown in Figure 3A, the expression of cytosolic superoxide dismutase (SOD) was upregulated by the loss of *kdmA* or *kdmB*. In contrast, expression levels of catalase, glutaredoxin Grx1, Mn SOD, and thioredoxin reductase were downregulated in both strains, and quinone oxidoreductase was significantly upregulated in Δ*kdmB* strain. In order to verify the RNA-seq results of these genes, RT-qPCR analysis was performed, and it was found that the expression of catalase, MnSOD, and thioredoxin reductase was decreased. The expression level of quinone oxidoreductase was increased about 3-fold in Δ*kdmB* strain (Appendix A). In order to confirm this, we incubated the conidia of WT, Δ*kdmA*, and Δ*kdmB* strains with solid MMY containing H_2_O_2_ or PQ. In both mutant strains, tolerance to H_2_O_2_ was significantly decreased. In addition, whereas the Δ*kdmA* strain showed decreased tolerance to PQ, the tolerance of the Δ*kdmB* mutant was significantly increased (Figure 3B). Furthermore, the activities of both conidia-specific catalase (CatA, about 1.6-fold) and mycelia-specific catalase (Cat1) were decreased compared to those of the WT strain following the loss of *kdmA* or *kdmB* expression (Figure 3C). The activities of cytoplasmic Cu/ZnSOD (Sod1) and mitochondrial MnSOD (Sod2) [44] were decreased in the Δ*kdmA* mutant, whereas in the Δ*kdmB* strain, the activity of Sod2 was enhanced (Figure 3D), suggesting that the increased resistance of the Δ*kdmB* mutant to PQ might be associated with higher Sod2 activity.

### 3.4. KdmA and KdmB Regulate Sensitivity to 6-Azauracil

Cells with histone methylation site mutations are frequently sensitive to chromatin-targeting inhibitors. Therefore, we tested the sensitivity of Δ*kdmA* and Δ*kdmB* mutants to the guanine nucleotide synthesis inhibitor and transcription defect indicator 6-azauracil (6AU), ribonucleotide reductase inhibitor and DNA replication efficiency indicator hydroxyurea, and microtubule destabilizer thiabendazole. Δ*kdmA* and Δ*kdmB* mutants had similar sensitivity to hydroxyurea and thiabendazole, whereas the radial growth rates of both strains were significantly lower than that of the WT strain in the presence of 6AU (45.7% and 52.7%, respectively; Figure 4A). In order to investigate the mechanism of the response to 6AU in more detail, we analyzed expression levels of the *imd2* gene (encoding IMP dehydrogenase) and *sdt1* gene (encoding pyrimidine nucleotidase). The expression of these genes is known to affect the resistance to 6AU [45,46]. We found that expression levels of *imd2* and *sdt1* were significantly lower in both mutant strains than that in the WT strain (Figure 4B), and there was no significant differential expression without stress.

### 3.5. KdmA and KdmB Oppositely Affect Gliotoxin Synthesis

Comparative transcriptome analysis revealed that while cytochrome P450 monooxygenase and GliZ-like C6 transcription factor were upregulated in Δ*kdmA* strain, MFS transporter was upregulated by the loss of *kdmB* (Figure 5A, Appendix A). Gliotoxin (GT) is a mycotoxin that inhibits the host immune response to fungal infection. In order to test the roles of KdmA and KdmB in GT synthesis, GT production was measured in the culture medium of each strain. Examination of the extracts of the two mutants showed that the Δ*kdmA* strain had increased GT production, whereas the Δ*kdmB* strain had decreased GT production compared to that in the WT strain (Figure 5B). The production of several unidentified metabolites was also decreased following *kdmB* deletion. In addition to having opposite effects on GT production, the mutations in *kdmA* and *kdmB* oppositely regulated expression levels of *laeA*, a regulator of secondary metabolism, and *gliZ*, a C6 transcription factor required for *gli* gene expression. The RT-qPCR results of *gliZ* in both mutants were consistent with RNA-seq findings (Figure 5C).

### 3.6. Virulence of the ΔkdmA and ΔkdmB Strains in Immunocompromised Mice

In order to assess the pathological significance of KdmA and KdmB, we examined how the deletion of their encoding genes affected the virulence of *A. fumigatus* in a murine model. The conidia of the WT and two mutant strains were introduced to neutropenic mice intranasally, and pathological outcomes were analyzed by monitoring mouse survival. Mice infected with the Δ*kdmA* mutant showed similar lethality to mice infected with the WT strain. In contrast, the virulence of the Δ*kdmB* mutant was greatly decreased (*p* = 0.036) (Figure 6A). In order to understand the basis for the differences in mouse survival, lung tissue sections were prepared from infected mice 3 days after inoculation and stained with hematoxylin and eosin (H&E) or periodic acid–Schiff (PAS). As shown in Figure 6B, conidia of the Δ*kdmA* strain caused more severe tissue damage and hyphal growth than those observed in mice exposed to the WT strain. However, in sections of the lungs infected with the Δ*kdmB* strain, H&E and PAS staining revealed a rather small number of fungal cells, which was about 4-fold lower than that in mice infected with the WT strain (Figure 6C). Overall, the Δ*kdmB* strain showed lower growth within the host niche might be due to lower adhesion and/or internalization of conidia.

### 3.7. Transcriptome Analysis

To investigate the roles of KdmA and KdmB in *A. fumigatus* biology, we carried out RNA-sequencing analysis using the mutants and WT cells. Of the 9859 genes, 3740 genes showed more than two-fold differential expression (*p* < 0.05), of which 1948 genes exhibited higher transcript levels in Δ*kdmA* strain than that in WT strain and 1784 genes were downregulated. In Δ*kdmB* strain, 3623 genes were differentially expressed more than two-fold (*p* < 0.05) and 1912 genes were upregulated, and 1640 genes were downregulated (Figure 7A, Appendix A). Functional category analysis was carried out by determining Gene Ontology (GO) terms that were enriched in differentially expressed genes (DEGs). The top significant molecular function GO category of Δ*kdmA* and Δ*kdmB* strain is “transporter activity” and “antioxidant activity”, respectively. The top significant cellular component GO category of both strains is “fungal-type cell wall”, and the top significant biological process GO category is “cell wall biogenesis”. The most enriched molecular function, cellular component, and biological process GO categories of both mutants are “catalytic activity”, “intrinsic component of membrane”, and “metabolic process”, respectively (Figure 7B).

## 4. Discussion

Post-translational modifications of histones involved in chromatin remodeling affect various biological processes through the regulation of gene cluster expression patterns. One such modification is the methylation of lysine residues of histone H3, which is regulated by two groups of proteins, methyl transferases, which methylate specific lysine residues, and demethylases, which remove the methylation marks. The methylation of histone lysine residues causes activation or repression of gene transcription, depending on their position and methylation state [47].

In the present study, we demonstrated how KdmA and KdmB affect the growth, development, and pathogenic features of *A. fumigatus*. Significantly reduced radial growth, conidiation, and mRNA levels of key asexual development regulators were observed in the Δ*kdmA* strain. In contrast, vegetative growth, conidiation, and transcript levels of asexual development regulators were increased by the loss of *kdmB*, suggesting that KdmA and KdmB regulate mycelial growth and asexual development in opposite ways. From these results, we can assume that although these demethylases have different or even opposite roles in the same pathways, they both might be required for the fine regulation of normal growth and proper asexual development of *A. fumigatus*.

Previously, it was reported that the loss of *kdmA* resulted in a greater sensitivity to oxidants such as H_2_O_2_ and PQ in *A. nidulans* [7]. We also observed that the Δ*kdmA* but not the Δ*kdmB* strain was more sensitive to H_2_O_2_ and PQ, suggesting that KdmB may regulate responses to oxidative stress through alternative mechanisms. Previous publications have demonstrated that H3K4 methylation is involved in the maintenance of resistance to the antifungal drug brefeldin A and a clinically used azole drug [48,49].

To determine whether KdmA and KdmB are involved in antifungal drug resistance, we performed E-strip tests with amphotericin B, flucytosine, and voriconazole. The sensitivity of the ∆*kdmB* mutants to amphotericin B was increased significantly, and both mutants displayed hypersensitivity to voriconazole. In spite of the active condition (pH 5.0) [50], flucytosine did not form a clear inhibition zone in all strains tested (Appendix A).

Interestingly, we found that both mutants showed increased sensitivity to 6AU, an inhibitor of GMP biosynthesis that was used to select mutants with defective transcriptional elongation [51,52,53]. This result was similar to that reported for the histone methyltransferase mutants, suggesting that proper regulation of histone methylation is required for GMP synthesis and/or transcriptional elongation activity [18,54,55]. Mechanistically, 6AU inhibits IMP dehydrogenase (IMD2/PUR5), an enzyme that catalyzes the first step of the GMP synthesis pathway [46], and SDT1/SSM1, a pyrimidine nucleotidase that contributes to the resistance to the toxic effect of 6AU [45]. We also observed that deletion of *kdmA* or *kdmB* significantly reduced IMP dehydrogenase (*imd2*) and pyrimidine nucleotidase (*sdt1*) transcript levels compared to those in the WT strain. However, unlike in the Δ*AfclrD* mutant with a deletion of H3K9 methyltransferase, which was sensitive to 6AU independently of changes in *imd2* and/or *sdt2* mRNA expression [18], the sensitivity of the Δ*kdmA* and Δ*kdmB* mutants to 6AU was proportional to changes in *imd2* and/or *sdt2* expression levels, indicating that the mechanisms of resistance to 6AU in the Δ*AfclrD*, Δ*kdmA*, and Δ*kdmB* mutants were quite different.

Although KdmA and KdmB of *A. nidulans* were shown to affect the expression of secondary metabolism-related genes [7,12], the roles of histone demethylases in the production of individual toxins are much less clear. In the present study, we found that KdmA and KdmB regulate the production of GT and the expression of genes involved in GT biosynthesis. To the best of our knowledge, this is the first report on the effects of histone demethylases on GT biosynthesis. Our results showed that GT production was markedly enhanced in *the* Δ*kdmA* mutant but severely decreased in the Δ*kdmB* strain compared with that in the WT strain. Regulation of GT biosynthesis involves numerous transcription factors, protein kinases, transcriptional and developmental regulators, regulators of G protein signaling, as well as chromatin-modifying enzymes [56]. Although the expression level *brlA* positively regulating the production of GT [22] was down-expressed, the levels of mRNA transcripts of the regulator genes *laeA* and *gliZ* were also altered in the mutants in the same fashion. The mortality rate of mice infected with the Δ*kdmB* mutant was lower than the mortality rates of animals infected with the WT or Δ*kdmA* mutant strains. Histological analysis revealed neither spore germination nor the invasion of hyphae in the lungs of mice infected with the Δ*kdmB* strain, suggesting that KdmB might be a key factor in fungal virulence. Further studies are needed to elucidate the pathological properties of the Δ*kdmA* strain.

Comparative transcriptomic analysis reveals both mutants downregulate the expression of developmental process-related genes and antioxidant activity-related genes, suggesting that both KdmA and KdmB are needed for the proper development and regulation of responses to external oxidative stress. As shown in Appendix A, the majority of deregulated genes connected to the GO term “primary metabolic process” were down-regulated in both mutants. From this result, we proposed that KdmA and KdmB may act as transcriptional repressors of primary metabolism genes similar to KdmA of *A. nidulans* [7].

In summary, our experiments with deletion mutants showed that KdmA and KdmB regulate vegetative growth, asexual sporulation, and GT production in opposite ways. However, both of these proteins were found to be involved in maintaining resistance to the toxic effects of 6AU.

## Figures and Tables

**Figure 1 jof-08-00590-f001:**
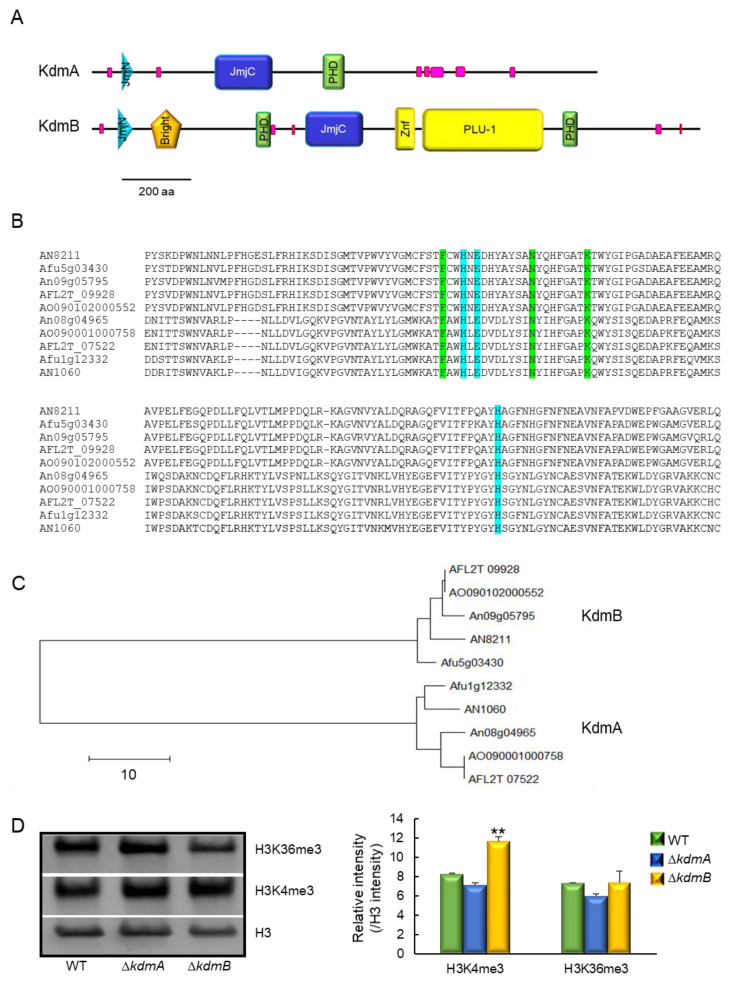
Molecular structures of KdmA and KdmB. (**A**) Schematic presentation of the domain structures of the KdmA and KdmB proteins using SMART (http://smart.embl-heidelberg.de, accessed on 4 April 2021). (**B**) Multiple sequence alignment of the JmjC domains of demethylases from *A. nidulans* (AN), *A. fumigatus* (Afu), *A. niger* (An), *A. flavus* (AFL), and *A. oryzae* (AO). Conserved residues responsible for α-ketoglutarate and Fe (II) binding sites are marked in green and blue, respectively. (**C**) A phylogenetic tree of demethylase-like proteins in various *Aspergillus* species constructed based on the matrix of neighbor-joining distances between the JmjC domain sequences. (**D**) Western blot with antibody specific to histone H3, H3K4me3, and H3K36me3 antibodies. Relative intensities are shown below. Statistical significance of differences was assessed by Student’s *t*-test: ** *p* < 0.01.

**Figure 2 jof-08-00590-f002:**
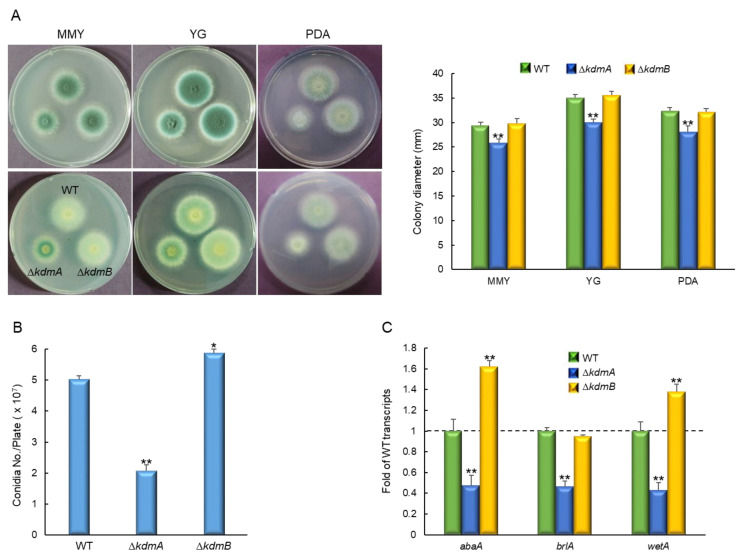
KdmA and KdmB regulate normal growth and asexual development in different ways. (**A**) Colony photographs and growth of WT, Δ*kdmA*, and Δ*kdmB* strains point-inoculated on various solid media and grown in solid glucose minimal medium with 0.1% yeast extract (MMY), 10 g/L yeast extract + 30 g/L glucose (YG), or potato dextrose agar (PDA) for 3 days and determined colony diameter. (**B**) Conidia numbers produced by each strain per plate. (**C**) Transcript levels of the key asexual developmental regulators in the mutants relative to the corresponding level in the WT strain at 3 days determined by quantitative RT-PCR (RT-qPCR). Dot line indicates the level of WT transcript. Fungal cultures were grown in MMY, and mRNA levels were normalized to the expression level of the *ef1α* gene. Statistical significance of differences was assessed by Student’s *t*-test: * *p* < 0.05, ** *p* < 0.01.

**Figure 3 jof-08-00590-f003:**
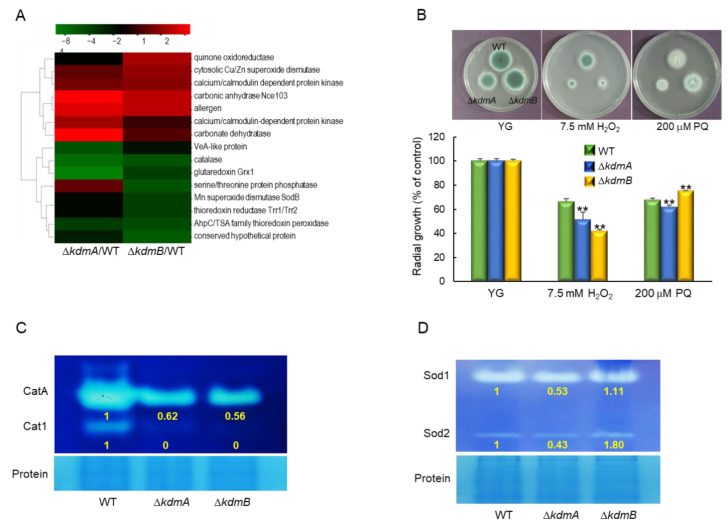
Distinct responses of Δ*kdmA* and Δ*kdmB* mutants to various kinds of oxidative stress. (**A**) Heatmap of altered expression levels of genes encoding oxidative stress-related proteins in Δ*kdmA* and Δ*kdmB* mutants. (**B**) Colony appearance and radial growth inhibition after inoculation of 1 × 10^5^ conidia on solid YG medium containing oxidative stressors. (**C**) Catalase activity of the WT and mutant strains. (**D**) SOD activity of the WT and mutant strains shown in non-denaturing polyacrylamide gels. Relative intensities of each enzyme’s activity are shown below. Statistical significance of differences between WT and mutant strains was evaluated using Student’s *t*-test: ** *p* < 0.01.

**Figure 4 jof-08-00590-f004:**
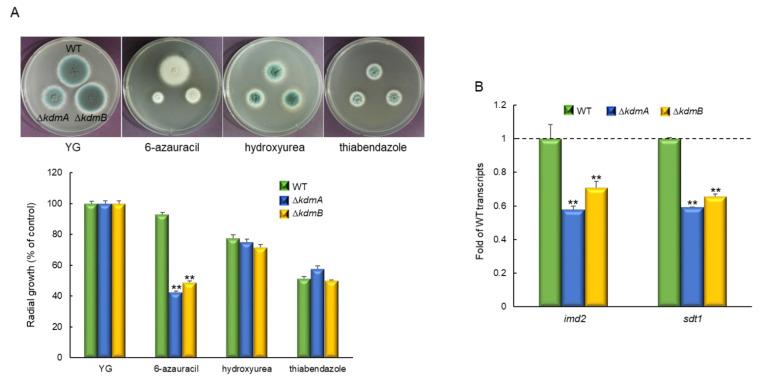
Increased sensitivity of the Δ*kdmA* and Δ*kdmB* mutants to 6AU. (**A**) Radial growth of the WT and mutant strains in the presence of chromatin-targeting inhibitors. (**B**) RT-qPCR analysis of expression levels of the 6AU resistance-related genes *imd2* and *sdt2* in the WT and mutant strains in the presence of 100 μg/mL of AU. Dot line indicates the level of WT transcript. Statistical significance of differences between WT and mutant strains was evaluated using Student’s *t*-test: ** *p* < 0.01.

**Figure 5 jof-08-00590-f005:**
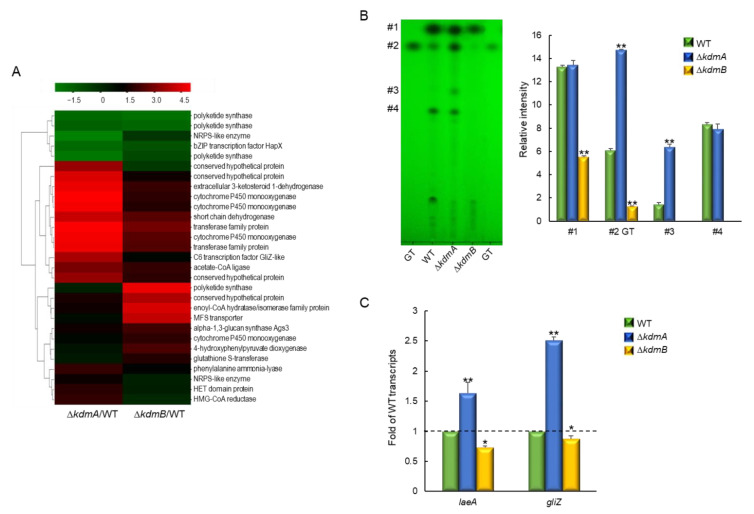
Roles of KdmA and KdmB in the production of gliotoxin (GT) and other putative secondary metabolites. (**A**) Heatmap of altered expression levels of genes encoding secondary metabolite biosynthesis in mutant strains. (**B**) Determination of GT production and several secondary metabolites in the WT and mutant strains. Standard GT concentration was 10 μg (left side) and 5 μg (right side). Left: a representative thin-layer chromatogram of the culture supernatant of each strain extracted with chloroform. Right: a graph of relative intensities of individual chromatogram bands in culture supernatants from different strains. (**C**) RT-qPCR analysis of changes the *laeA* and *gliZ* gene expression levels in mutant strains compared to that in the WT strain. Dot line indicates the level of WT transcript. Statistical significance of differences between WT and mutant strains was assessed by Student’s *t*-test: * *p* < 0.05, ** *p* < 0.01.

**Figure 6 jof-08-00590-f006:**
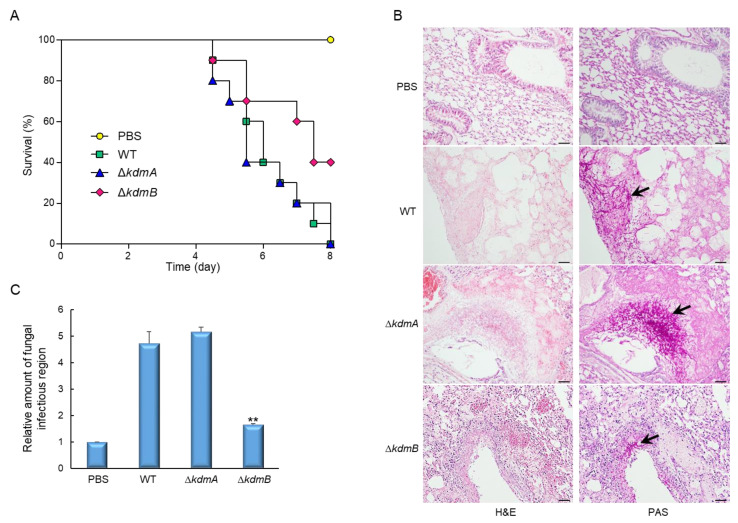
Effects of KdmA and KdmB on *A. fumigatus* virulence. (**A**) Survival curves of mice intranasally administered with PBS or with conidia of the WT or one of the mutant strains (n = 10/group). (**B**) Representative lung sections of mice from different experimental groups stained with hematoxylin and eosin (H&E) or periodic acid–Schiff reagent (PAS). Arrows indicate fungal mycelium. Scale bar = 50 µm. (**C**) Fungal burden in the lungs of mice infected with the WT or one of the mutant strains. Statistical significance of differences between WT and mutant strains was evaluated by Student’s *t*-test: ** *p* < 0.01.

**Figure 7 jof-08-00590-f007:**
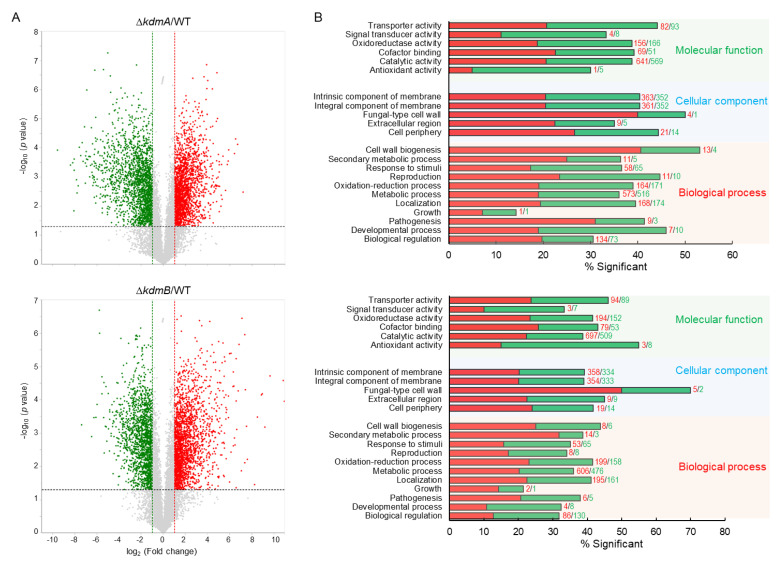
Genome-wide expression analyses of the Δ*kdmA* and Δ*kdmB*. (**A**) Volcano plot showing the fold change (x-axis) and *p*-value (y-axis) of genes sequenced in Δ*kdmA* (upper) and Δ*kdmB* (lower). Red and green dots denote up- and downregulated genes, respectively. (**B**) Functional annotation histograms of DEGs in Δ*kdmA* (upper) and Δ*kdmB* (lower). The red bars represent genes whose mRNA levels increased in the mutant, whereas the green bars represent those genes whose mRNA levels decreased in the mutant strain.

**Table 1 jof-08-00590-t001:** Strains used in this study.

Strain	Genotype	Reference
*A. nidulans* FGSC4	*veA*^+^ (Wild type)	FGSC ^a^
Af293	Wild type	[16]
Af293.1	*pyrG1*	[19]
Δ*kdm**A*	Δ*kdmA*::*AnipyrG*; *AfupyrG1*	This study
Δ*kdmB*	Δ*kdmB*::*AnipyrG*; *AfupyrG1*	This study
C’ *kdmA*	Δ*kdmA*::*AnipyrG*; *AfupyrG1*; *kdmA*::*hygB*	This study
C’ *kdmB*	Δ*kdmB*::*AnipyrG*; *AfupyrG1*; *kdmB*::*hygB*	This study

^a^ FGSC, Fungal Genetics Stock Center.

## Data Availability

RNA-seq data are available from the NCBI Gene Expression Omnibus (GEO) database (GSE166061).

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
