# Peer review of "The Lysine Demethylases KdmA and KdmB Differently Regulate Asexual Development, Stress Response, and Virulence in Aspergillus fumigatus"

_jof, 2022, doi:10.3390/jof8060590_

Round 1

Reviewer 1 Report

The manuscript by Choi et al. reported that two putative histone demethylases, KdmA and KdmB were involved in asexual development, drug resistance, gliotoxin production and virulence by erasing methylation mark on specific histone tail lysines in A. fumigatus. The authors accomplish this by combining loss of function and complementation experiments. In order to understand the regulatory network of KdmA and KdmB, they performed RNA-sequencing analysis in the kdmA and kdmB null mutants and found 3740 DEGs in ∆kdmA and 3623 DEGs in ∆kdmB mutant, respectively, and revealed that the expression levels of developmental process related genes and antioxidant activity related genes were downregulated in both mutants, which are consistent with the phenotypic changes. The experimental approaches used by the authors are appropriate and data provided are in good quality. In my opinion, this manuscript is worth to publish, but still I have some suggestions.

Minor concerns:

  1. Line 14. “Aspergillus fumigatus” should be in italic.
  2. Line 15, 16, 21. “∆AfukdmA” and “∆AfukdmB”, the genes name should be in italic.
  3. Line 55. “Jumonji histone demethylase proteins that demethylates H3K9 and H3K36.” Please list the respective reference(s).
  4. Line 211. What is the cultured condition of the indicated strains used for Western blotting?
  5. Line 247. Why do the authors use paraquat (PQ) here?
  6. Line 342. Both KdmA and KdmB are histone demethylases, the authors had determined the DEGs both in ∆kdmA and ∆kdmB mutant, it would be better if they could compare the common or different DEGs in these two mutants and link to their different phenotypes. It would be interesting to check the relative expression of biosynthetic gene clusters in both mutants.
  7. Line 381-382. Please list the respective reference(s).
  8. Line 400. Delete one “in”.

Reviewer 2 Report

This manuscript describes the deletion of two putative histone demethylase genes from three levels: fungal development, stress response, and virulence. This study is a complement for the histone demethylase studies in filamentous fungi. Although KdmA and KdmB has been identified and characterized in A. nidulans, the authors provides further evidences of their pleiotropic functions and mechanisms in human pathogen A. fumigatus.
However, there are some concerns for the authors to consider:

1. Line 227 – “the colony diameter of AfukdmB was slightly increased in all culture media (Figure 2A)”. The statistical result of colony diameter from PDA is contradictory to this conclusion.

2. Line 306 – 307 BrlA is reported to positively regulate gliotoxin production. Here, brlA was down expressed while GT production was increased in ΔAfukdmA. Please at least provide an explanation or discussion in the text for this result.

3. Figure 7 –Did you run a separate experiment for the histology study? The procedures for care and use of animals were approved by the Ethics Committee? Please provide P value of statistical analysis for the survival study. The overall survival curves between WT and ΔkdmA groups appears insignificant. This is the major concern in this article.

4. The description and conclusions regarding transcriptome data is inadequate. You can perform a heat map analysis for some annotated genes involving in oxidative stress response and gliotoxin synthesis.

Line 61 – the full name of the word 'JARID'
Line 67 – “but also as a positive regulator” I cannot conclude here.
Line 211 –Why only chose H3K4 and H3K36 as substrates of KdmA and KdmB?
Line 223 –lower-case “k” 
Line 274–“expression of these genes is known to affect the resistance to 6AU”. Reference is missing.
Line 275, 295, 333, 346–than that in 
Line 287-288 –Differences in MIC are considered when changes are greater than two fold and this would be no clinical significant.
Line 295 –Figure 5C
Line 305 –roles
Line 400 –in in?
Line 401 –were also
Line 403 –by the presence by?
Line 413 –individual toxin
Line 432 –both of

Reviewer 3 Report

In the manuscript entitled "The Putative Histone Demethylases, KdmA  and KdmB Differently Regulate Asexual Development, Stress Response, and Virulence in Aspergillus fumigatus" by Choi et al., the authors present a characterization of two lysine demethylase mutant strains, ∆kdmA and ∆kdmB, in the fungal mold pathogen Aspergillus fumigatus. Research on enzymes involved in the regulation of chromatin structure and post-translational modification in general is important in filamentous fungi, since already numerous of those enzymes have been reported to play a role for virulence of these organisms. In general, data generation seems to be sound, however, some results lack a detailed description of how they have been obtained. Moreover, some conclusions drawn by the authors are not sufficiently supported by the presented data. Therefore, major revision is recommended before accepting this manuscript for publication.

Major points:

Fig.1D:
I cannot follow the authors' claim that "AfuKdmA demethylates H3K4 and H3K36, but AfuKdmB demethylates only H3K4" (lines 211-12). Taking into account the intensities of H3, only H3K4 seems to be slightly increased in ∆kdmB, whereas both marks are reduced in ∆kdmA. What do the authors mean by "relative intensities"? Authors should clarify this. Normalized results should preferably be presented as columns, as has been performed for the growth assays. Also, should these results be discussed with regard to those found in Aspergillus nidulans. It would not be surprising if differences in the abundance of different methylation marks could not be detected on bulk histone level. I also would not doubt that these enzymes have different histone lysine residues as substrates and therefore are histone demethylases. Nevertheless, I strongly suggest to refer to them as lysine demethylases, as their gene names suggest. 

Fig.2:
From the colonies shown in panel A and also from the colony diameter measurements, there does not seem to be a significant increase in radial growth of ∆kdmB. However, in Fig.S1, these data look more convincing. Therefore, authors should show more representative colonies in the main figure.

Regarding the conidiation: Which medium was used for the conidiation assay? Can the authors exclude that the reduced growth phenotype of ∆kdmA does not interfere with their setup? Interestingly, on MMY and YG media, conidiation below the surface even seems to be increased in ∆kdmA when compared to wt and ∆kdmB.   

Fig.3B/C:
A loading control, like a Coomassie-stained gel, is missing. 
How much H2O2 and PQ, respectively, were added to the cultures?

Fig.4:
Which medium was used?
Was the transcription analysis performed at challenging conditions? If not, how would mutant strains respond to 6-AU compared to wild type?

Fig.5:
An enlarged view of the intersection area of the Etest strips should be provided, so that the concentration scale is readable. 
Whereas a clear difference of susceptibility to flucytosine of both mutants can be seen,
this is not the case for voriconazole. The observed MIC difference is rather small and I wonder if this can be regarded as significant, compare PMID 31766762: "In most of the studies, a threshold at +/−2 log2 dilutions is used".
In order to confirm differences in voriconazole susceptibility, a statistical analysis of the voriconazole MIC would be necessary. 
Regarding expression of cyp51 genes, analysis under challenging conditions would be informative. From the Etest strips I would not expect impairment of the transcriptional response to azole stress in the kdm mutants.     
Why are amphotericin B data not shown?
Due to the pH-dependence of flucytosine efficacy, the medium and the respective pH used for these analyses should be disclosed.
Further, authors should conduct flucytosine susceptibility testing in buffered media at pH 5 and pH 7 to get a hint if pH sensing would be perturbed in the kdm mutants.
PacC is not a pH-regulating protein, but a pH-dependent transcription factor.   

Fig.6:
The amount of GT loaded as control should be shown. Has the same amount been loaded on both sides of the chromatogram?

Fig.7:
Although ∆kdmB seems to display virulence attenuation after 8 days of infection, it could just be delayed for 1–2 days. Has survival been assessed for additional days? The p-value of the log-rank test of the survival curves should be disclosed. 
The Immunosuppression should be described in more detail in the materials and methods section, how much of each substance and when it was applied.
It should be explained how fungal burden was determined. Authors should discuss the discrepancy between histological and survival data. 
It seems like ∆kdmB grows even better on rich media (Fig.S1) but shows a growth defect in the host niche. 
Authors should conduct growth assays on 10–20% blood agar plates, to assess if reduced growth of ∆kdmB in the lung is caused by nutrient depletion. 

The discussion section should be revised.
To a large extent the discussion is a repetition of the results. Even obviously puzzling results within the manuscript are not discussed; e.g., seemingly lack of growth of ∆kdmB after 3 days in the lung vs. only slight attenuation of virulence.
The authors state "the pattern of changes of GT production correlated with virulence in immunocompromised mice". However, since the immunosuppression applied leads to neutropenia, it is rather unlikely that GT contributes significantly to virulence attenuation because neutrophils are the main target of GT (e.g., PMID 17601876). 
Possibly, in a non-neutropenic infection model the virulence phenotype would be more pronounced.  

The RNAseq analysis is discussed only poorly. Can the differential regulation of the genes presented via RT-qPCR analysis also be found in the RNAseq data? Since transcriptome analysis has also been performed with ∆kdmA and ∆kdmB mutants in A. nidulans, I encourage authors to compare differential expression of kdm mutants between A. fumigatus and A. nidulans.
Thin layer chromatography revealed additional differences apart from GT, it therefore might be worth to check RNAseq data for differential expression of SM gene clusters. Finally, authors should present the results for all genes of their RNAseq experiment as a spreadsheet in the supplementary data. 

Minor points:
Line 2: Omit first comma in the article title
Line 14: For the sake of readability I suggest to omit "Afu" when referring to demethylase genes/enzymes because it is already clear from the article title that the focus is on Aspergillus fumigatus. However, when referring to non-Afu genes/proteins, this should be clarified.    
Line 85: “Bioinformatic Analysis…”
Line 87: Af293. Please correct all occurrences.
Line 100: A. fumigates should be removed.
Line 135: Paraquat (PQ) should be defined already here.
Line 153: Could it be that higher percentage gels (14–16%) have been used to separate nuclear extracts?
Line 173: Have RNAseq results been confirmed for all genes?
Line 177: Mice with average body weight of 30 g?
Line 251: Omit "p" after protein symbols throughout the article when referring to Afu proteins. Generally, please make sure to stick to nomenclature of Aspergillus genes and proteins.
Line 272–73: imd2 and sdt1 are yeast genes? Please clarify that expression of the respective Afu orthologs was determined in text and figure.
Line 314: "and OTHER PUTATIVE secondary metabolites"
Line 342: Transcriptome
Line 344: "probes" should read genes

Round 2

Reviewer 2 Report

The authors have been very responsive to previous critiques. There are still some (English) minor issues.

1. Line 422–Although the expression level of brlA positively regulating the production of GT [22] was down-expressed, the levels of mRNA transcripts of the regulator genes laeA and gliZ were also altered in the mutants in the same fashion (Figure 6)”

2. Line 328–…genes encoding secondary metabolite biosynthesis…

3. Line 324–The RT-qPCR results of gliZ in both mutants were consistent with the RNA-seq findings.

4. Line 271-272–…and it was found that…

Just as mentioned above, check the whole manuscript in detail.

5. For the review “Line 287-288 –Differences in MIC are considered when changes are greater than two fold and this would be no clinical significant”, what I meant to say is that sensitivity of the ∆kdmA and ∆kdmB mutants to voriconazole was not increased significantly. Interestingly, both mutants displayed hypersensitivity to flucytosine.

6. For the review "Line 211 –Why only chose H3K4 and H3K36 as substrates of KdmA and KdmB?", the question is, for example, if H3K9 is a substrate of KdmA or KdmB?

Reviewer 3 Report

Thank you for the amendments made.

When needed, I used your response file to introduce my comments.
